# Modelling and unsupervised learning of symmetric deformable object categories

**James Thewlis**[1]  **Hakan Bilen**[2]  **Andrea Vedaldi**[1]

[1] Visual Geometry Group
University of Oxford
`{jdt,vedaldi}@robots.ox.ac.uk`

[2] School of Informatics
University of Edinburgh
`hbilen@ed.ac.uk`

## Abstract

We propose a new approach to model and learn, without manual supervision, the symmetries of natural objects, such as faces or flowers, given only images as input. It is well known that objects that have a symmetric structure do not usually result in symmetric images due to articulation and perspective effects. This is often tackled by seeking the intrinsic symmetries of the underlying 3D shape, which is very difficult to do when the latter cannot be recovered reliably from data. We show that, if only raw images are given, it is possible to look instead for symmetries in the *space of object deformations*. We can then learn symmetries from an unstructured collection of images of the object as an extension of the recently-introduced *object frame* representation, modified so that object symmetries reduce to the obvious symmetry groups in the normalized space. We also show that our formulation provides an explanation of the ambiguities that arise in recovering the pose of symmetric objects from their shape or images and we provide a way of discounting such ambiguities in learning.

## 1 Introduction

Most natural objects are symmetric: mammals have a bilateral symmetry, a glass is rotationally symmetric, many flowers have a radial symmetry, etc. While such symmetries are easy to understand for a human, it remains surprisingly challenging to develop algorithms that can reliably detect the symmetries of visual object in images. The key difficulty is that objects that are structurally symmetric do not generally result in symmetric images; in fact, the latter occurs only when the object is imaged under special viewpoints and, for deformable objects, with a special poses (Leonardo's Vitruvian Man illustrates this point).

The standard approach to characterizing symmetries in objects is to look not at their images, but at their 3D shape; if the latter is available, then symmetries can be recovered by analysing the *intrinsic geometry* of the shape. However, often only images of the objects are available, and reconstructing an accurate 3D shape from them can be very challenging, especially if the object is deformable.

In this paper, we thus seek a new approach to learn *without supervision and from raw images alone* the symmetries of deformable object categories. This may sound difficult since even characterising the basic geometry of natural objects without external supervision remains largely an open problem. Nevertheless, we show that it is possible to extend the method of [38], which was recently introduced to learn the "topology" of object categories, to do exactly this.

There are three key enabling factors in our approach. First, we do not consider symmetries of a single object or 3D shape in isolation; instead, we seek symmetries shared by all the instances of the objects in a given category, imaged under different viewing conditions and deformations. Second, rather than considering the common concept of intrinsic symmetries, we propose to look at symmetries not of 3D shapes, but of the *space of their deformations* (section 4). Third, we show that the *normalized object frame* of [38] can be learned in such a way that the deformation symmetries are represented by

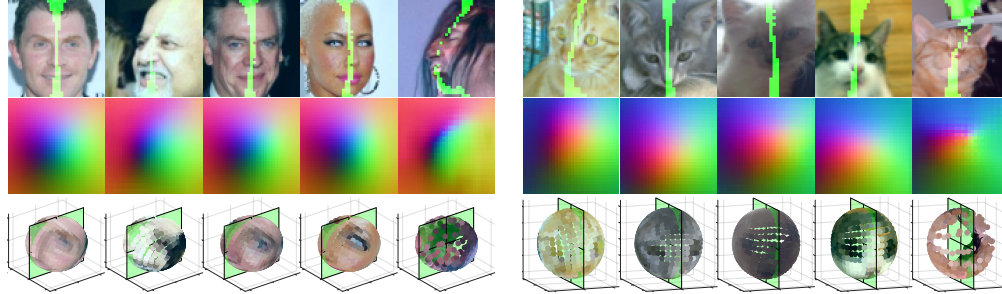

Figure 1: **Symmetric object frame for human (left) and cat (right) faces** (test set). Our method learns a viewpoint and identity invariant geometric embedding which captures the symmetry of natural objects (in this case bilateral) *without manual supervision*. Top: input images with the axis of symmetry superimposed (shown in green). Middle: dense embedding mapped to colours. Bottom: image pixels mapped to 3D representation space with the reflection plane (green).

the obvious symmetry groups in the object frame. The latter also result in a constraint that can be easily added to the self-supervised formulation of [38] to learn symmetries in practice (section 3).

We start by deriving our formulation for the special case of bilateral symmetries (section 3). Then, we propose a theory of symmetric deformation spaces (section 4) that generalises the method to other symmetry groups. An important step in this generalization is to characterise the ambiguities that symmetries induce in recovering the pose of an object from an image of it, or from its 3D shape, which may not occur with bilateral symmetries.

The resulting approach is the first that, to our knowledge, can learn the symmetries of object categories given only raw images as input, without manual annotations. For demonstration, we show that this approach can learn the bilateral symmetry in human and pet faces (fig. 1) as well as in synthetic 3D objects (section 6). To assess the method, we look at how well the resulting representation can detect pairs of symmetric object landmarks (e.g. left and right eyes) even when the object does not appear symmetric.

We also investigate the problem of symmetry-induced ambiguities in learning the geometry of natural objects. For objects such as animals that have a bilateral symmetry, it is generally possible to uniquely identify their left and right sides and thus recover their pose uniquely. On the other hand, for objects such as flowers that may have a radial symmetry, it is generally impossible to say which way is "up", creating an ambiguity in pose recovery. Our framework clarifies why and when this occurs and suggests how to modify the learning formulation to mitigate the effect of such ambiguities (sections 4 and 6.2).

## 2   Related work

**Cross-instance object matching.**   Our method is also related to the techniques that find dense correspondences between different object instances by matching their SIFT features [25], establishing region correspondences [14, 15] and matching the internal representations of neural networks [24]. In addition, dense correspondences have been generalized between image pairs to arbitrary number of multiple images by Learned-Miller [20]. More recently, RSA [32], Collection Flow [18] and Mobahi *et al*. [28] show that a collection of images can be projected into a lower dimensional subspace before performing a joint alignment among the projected images. Novotny *et al*. [30] train a neural network with image labels that learns to automatically discover semantically meaningful parts across animals.

**Unsupervised learning of object structure.**   Supervised visual object characterization [6, 11, 21, 8, 10] is a well established problem in computer vision and successfully applied to facial landmark detection and human body pose estimation. Unsupervised methods include Spatial Transformer Networks [16] that learn to transform images to improve image classification, WarpNet [17] and geometric matching networks [34] that learn to match object pairs by estimating relative transformations between them. In contrast to ours, these methods do not learn a canonical object geometry and only provide relative mapping from one object to another. More related to ours, Thewlis *et al*. [39, 38] propose to characterize object structure via detecting landmarks [39] or dense labels [38] that are

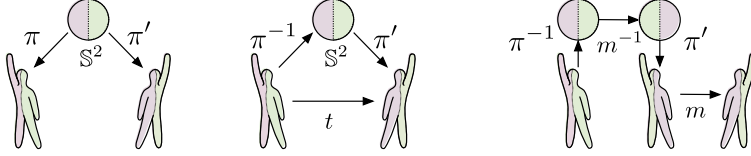

Figure 2: Left: an object category consisting of two poses $\pi, \pi'$ with bilateral symmetry. Middle: the non-rigid deformation $t = \pi' \circ \pi^{-1}$ transporting one pose into the other. Right: construction of $t = m\pi m^{-1}\pi^{-1}$ by applying the reflection operator $m$ both in Euclidean space and in representation space $\mathbb{S}^2$. This also shows that the symmetric pose $\pi' = m\pi m^{-1}$ is the "conjugate" of $\pi$.

consistent with object deformations and viewpoint changes. In fact, our method builds on [38] and also learns a dense geometric embedding for objects, however, by using a different supervision principle, symmetry.

**Symmetry.** Computational symmetry [22] has a long history in sciences and played an essential role in several important discoveries including the theory of relativity [29], the double helix structure of DNA [42]. Symmetry is shown to help grouping [19] and recognition [41] in human perception. There is a vast body of computer vision literature dedicated to finding symmetries in images [26], two dimensional [1] and three dimensional shapes [37]. Other axes of variations among symmetry detection methods are whether we seek transformations to map the whole [33] or part of an object [12] to itself; whether distances are measured in the extrinsic Euclidean space [1] or with respect to an intrinsic metric of the surface [33]. In addition to symmetry detection, symmetry is also used as prior information to improve object localization [4], text spotting [47], pose estimation [44] and 3D reconstruction [35]. Symmetry constraints been used to find objects in 3D point clouds [9, 40]. Symmetrization [27] can be used to warp meshes to a symmetric pose. Symmetry cues can be used in segmentation [3, 5]. [2] learns representations that respect a group structure learned from data symmetries.

## 3    Self-supervised learning of bilateral symmetries

In this section, we extend the approach of [38] to learn the bilateral symmetry of an object category.

**Object frame.** The key idea of [38] is to study 3D objects not via 3D reconstruction, which is challenging, but by characterizing the correspondences between different 3D shapes of the object, up to pose or intra-class variations.

In this model, an *object category* is a space $\Pi$ of homeomorphisms $\pi : \mathbb{S}^2 \to \mathbb{R}^3$ that embed the sphere $\mathbb{S}^2$ into $\mathbb{R}^3$. Each possible *shape* of the object is obtained as the (mathematical) image $S = \pi[\mathbb{S}^2]$ under a corresponding function $\pi \in \Pi$, which we therefore call a *pose* of the object (different poses may result in the same shape). The correspondences between a pair of shapes $S = \pi[\mathbb{S}^2]$ and $S' = \pi'[\mathbb{S}^2]$ is then given by $\pi' \circ \pi^{-1}$, which is a bijective deformation of $S$ into $S'$.

Next, we study how poses relate to images of the object. A (color) image is a function $\mathbf{x} : \Omega \to \mathbb{R}^3$ mapping pixels $u \in \Omega$ to colors $\mathbf{x}_u$. Suppose that $\mathbf{x}$ is the image of the object under pose $\pi$; then, a point $z \in \mathbb{S}^2$ on the sphere projects to a point $\pi z \in \mathbb{R}^3$ on the object surface $S$ and the latter projects to a pixel $u = \text{Proj}(\pi z) \in \Omega$, where $\text{Proj}$ is the camera projection operator.

The idea of [38] is to learn a function $\psi_u(\mathbf{x})$ that "reverses" this process and, given a pixel $u$ in image $\mathbf{x}$, recovers the corresponding point $z$ on the sphere (so that $\forall u : u = \text{Proj}(\pi\psi_u(\mathbf{x}))$). The intuition is that $z$ identifies a certain object landmark (e.g. the corner of the left eye in a face) and that the function $\psi_u(\mathbf{x})$ recovers which landmark lands at a certain pixel $u$.

The way the function $\psi_u(\mathbf{x})$ is learned is by considering pairs of images $\mathbf{x}$ and $\mathbf{x}' = t\mathbf{x}$ related by a *known* 2D deformation $t : \Omega \to \Omega$ (where the warped image $t\mathbf{x}$ is given by $(t\mathbf{x})_u = \mathbf{x}_{t^{-1}u}$). In this manner, pixels $u$ and $u' = tu$ are images of the *same* object landmark and therefore must project on the same sphere point. In formulas, and ignoring visibility effects and other complications, the learned function must satisfy the *invariance constraint*:

$$\forall u \in \Omega : \quad \psi_u(\mathbf{x}) = \psi_{tu}(t\mathbf{x}) \tag{1}$$

In practice, triplets $(\mathbf{x}, \mathbf{x}', t)$ are obtained by *randomly sampling* 2D warps $t$, assuming that the latter approximate warps that could arise form an actual pose change $\pi' \circ \pi^{-1}$. In this manner, knowledge of $t$ is automatic and the method can be used in an unsupervised setting.

**Symmetric object frame.** So far the object frame has been used to learn correspondences between different object poses; here, we show that it can be used to establish auto-correspondences in order to model object symmetries as well.

Consider in particular an object that has a *bilateral symmetry*. This symmetry is generated by a reflection operator, say the function $m : \mathbb{R}^3 \to \mathbb{R}^3$ that flips the first axis:

$$m : \qquad \mathbb{R}^3 \to \mathbb{R}^3, \qquad \begin{bmatrix} p_1 \\ p_2 \\ p_3 \end{bmatrix} \mapsto \begin{bmatrix} -p_1 \\ p_2 \\ p_3 \end{bmatrix}. \qquad (2)$$

If $S$ is a shape of a bilaterally-symmetric object, no matter how we align $S$ to the symmetry plane, in general $m[S] \neq S$ due to object deformations. However, we can expect $m[S]$ to still be a valid shape for the object. Consider the example of fig. 2 of a person with his/her right hand raised; if we apply $m$ to this shape, we obtain the shape of a person with the left hand raised, which is valid.

However, reasoning about shapes is insufficient to apply the object frame model; we require instead to work with correspondences, encoded by poses. Unfortunately, even though $m[S]$ is a valid shape, $m$ is *not* a valid correspondence as it flips the left and right sides of a person, which is not a "physical" deformation (why this is important will be clearer later; intuitively it is the reason why we can tell our left hand from the right by looking).

Our key intuition is that we can *learn* the pose representation in such a way that the correct correspondences are trivially expressible there. Namely, assume that $m$ applied to the sphere amounts to swapping each left landmark of the object with its corresponding right counterpart. The correct deformation $t$ that maps the "right arm raised" pose to the "left arm raised" pose can now be found by applying $m$ first in the normalized object frame (to swap left and right sides while leaving the shape unchanged) and then again in 3D space (undoing the swap while actually deforming the shape). This two-step process is visualised in fig. 2 right.

This derivation is captured by a simple change to constraint (1), encoding equivariance rather than invariance w.r.t. the warp $m$:

$$\forall u \in \Omega : \quad m\psi_u(\mathbf{x}) = \psi_{mu}(m\mathbf{x}) \qquad (3)$$

We will show that this simple variant of eq. (1) can be used to learn a representation of the bilateral symmetry of the object category.

**Learning formulation.** We follow [38] and learn the model $\psi_u(\mathbf{x})$ by considering a dataset of images $\mathbf{x}$ of a certain object category, modelling the function $\psi_u(\mathbf{x})$ by a convolutional neural network, and formulating learning as a Siamese configuration, combining constraints (3) and (1) into a single loss. To avoid learning the trivial solution where $\psi_u(\mathbf{x})$ is the constant function, the constraints are extended to capture not just invariance/equivariance but also distinctiveness (namely, equalities (3) and (1) should *not* hold if $u$ is replaced with a different pixel $v$ in the left-hand side). Following [38], this is captured probabilistically by the loss:

$$\mathcal{L}(\mathbf{x}, m, t) = \int_\Omega \|v - mtu\|_2^\gamma p(v|u) \, dv du, \quad p(v|u) = \frac{\exp\langle m\psi_u(\mathbf{x}), \psi_v(mt\mathbf{x})\rangle}{\int \exp\langle m\psi_u(\mathbf{x}), \psi_w(mt\mathbf{x})\rangle \, dw} \qquad (4)$$

The probability $p(v|u)$ represents the model's belief that pixel $u$ in image $\mathbf{x}$ matches pixel $v$ in image $mt\mathbf{x}$ based on the learned embedding function; the latter is relaxed to span $\mathbb{R}^3$ rather than only $\mathbb{S}^2$ to allow the length of the embedding vectors to encode the belief strength (as shorter vectors results in flatter distributions $p(v|u)$). For unsupervised training, warps $t \sim T$ are randomly sampled from a fixed distribution $T$ as in [38], whereas $m$ is set to be either the identity or the reflection along the first axis with 50% probability.

## 4 Theory

In the previous section, we have given a formulation for learning the bilateral symmetry of an object category, relying mostly on an intuitive derivation. In this section, we develop the underlying theory in a more rigorous manner (proofs can be found in the supplementary material), while clarifying three important points: how to model symmetries other than the bilateral one, why symmetries such as radial result in ambiguities in establishing correspondences and why this is usually not the case for the bilateral symmetry, and what can be done to handle such ambiguities in the learning formulation when they arise.

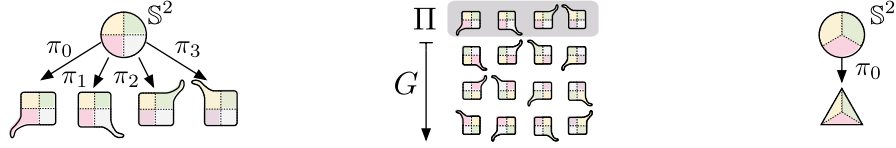

Figure 3: Left: a set $\Pi = \{\pi_0, \ldots, \pi_3\}$ of four poses with rotational symmetry group $H = \{h^k, k = 0, 1, 2, 3\}$ where $h$ is a rotation by $\pi/2$. Note that none of the shapes is symmetric; rather, the object, which stays "upright", can deform in four symmetric ways. The shape of the object is then sufficient to recover the pose uniquely. Middle: closure of the pose space $\Pi$ by rotations $G = H$. Now pose can be recovered from shapes only up to the symmetry group $H$. Right: an equilateral triangle is represented by a pose $\pi_0$ invariant to conjugation by 60 degrees rotations (which are the "ordinary" extrinsic symmetries of this object).

**Symmetric pose spaces.** A *symmetry* of a shape $S \subset \mathbb{R}^3$ is often defined as an isometry[1] $h : \mathbb{R}^3 \to \mathbb{R}^3$ that leaves the set invariant, i.e. $h[S] = S$. This definition is not very useful when dealing with symmetric but deformable objects, as it works only for special poses (cf. the Vitruvian Man); we require instead a definition of symmetry that is not pose dependent. A common approach is to define *intrinsic symmetries* [33] as maps $h : S \to S$ that preserve the geodesic distance $d_S$ defined on the surface of the object (i.e. $\forall p, q \in S : d_S(hp, hq) = d_S(p, q)$). This works because the geodesic distance captures the intrinsic geometry of the shape, which is pose invariant (but elastic shape deformations are still a problem); however, using this definition requires to accurately reconstruct the 3D shape of objects from images, which is very challenging.

In order to sidestep this difficulty, we propose to study the symmetry not of the 3D shapes of objects, but rather of the space of their deformations. As discussed in section 3, such deformations are captured as a whole by the pose space $\Pi$. We define the *symmetries* of the pose space $\Pi$ as the subset of linear isometries that leave $\Pi$ unchanged via conjugation:

$$H(\Pi) = \{h \in O(3) : \forall \pi \in \Pi : h\pi h^{-1} \in \Pi \land h^{-1}\pi h \in \Pi\}.$$

For example, in fig. 2 we have obtained the "left hand raised" pose $\pi'$ from the "right hand raised" pose via conjugation $\pi' = m\pi m^{-1}$ via the reflection $m$ (note that $m = m^{-1}$).

**Lemma 1.** *The set $H(\Pi)$ is a subgroup of $O(3)$.*

The symmetry group $H(\Pi)$ partitions $\Pi$ in equivalence classes of symmetric poses: two poses $\pi$ and $\pi'$ are symmetric, denoted $\pi \sim_{H(\Pi)} \pi'$, if, and only if, $\pi' = h\pi h^{-1}$ for an $h \in H(\Pi)$. In fact:

**Lemma 2.** $\pi \sim_{H(\Pi)} \pi'$ *is an equivalence relation on the space of poses $\Pi$.*

Figure 3 shows an example of an object $\Pi$ that has four rotationally-symmetric poses $H(\Pi) = \{h^k \pi_0 h^{-k}, k = 0, 1, 2, 3\}$ where $h$ is a clockwise rotation of 90 degrees.

**Motion-induced ambiguities.** In the example of fig. 3, the object is pinned at the origin of $\mathbb{R}^3$ and cannot rotate (it can only be "upright"); in order to allow it to move around, we can extend the pose space to $\Pi' = G\Pi$ by applying further transformations to the poses. For example, choosing $G = SE(3)$ to be the Euclidean group allows the object to move rigidly; fig. 3-middle shows an example in which $G = H(\Pi)$ is the same group of four rotations as before, so the object is still pinned at the origin but not necessarily upright.

Motions are important because they induce ambiguities in pose recover. We formalise this concept next. First, we note that, if $G$ contains $H(\Pi)$, extending $\Pi$ by $G$ preserves all the symmetries:

**Lemma 3.** *If $H(\Pi) \subset G$, then $H(\Pi) \subset H(G\Pi)$.*

Second, consider being given a shape $S$ (intended as a subset of $\mathbb{R}^3$) and being tasked with recovering the pose $\pi \in \Pi$ that generates $S = \pi[\mathbb{S}^2]$. Motions makes this recovery ambiguous:

**Lemma 4.** *Let the pose space $\Pi$ be closed under a transformation group $G$, in the sense that $G\Pi = \Pi$. Then, if pose $\pi \in \Pi$ is a solution of the equation $S = \pi[\mathbb{S}^2]$ and if $h \in H(\Pi) \cap G$, then $\pi h^{-1}$ is another pose that solves the same equation.*

Lemma 4 does not necessarily provide a complete characterization of all the ambiguities in identifying pose $\pi$ from shape $S$; rather, it captures the ambiguities arising from the symmetry of the object and its ability to move around in a certain manner. Nevertheless, it is possible for specific poses to result in further ambiguities (e.g. consider a pose that deforms an object into a sphere).

In order to use the lemma to characterise ambiguities in pose recovery, given a pose space $\Pi$ one must still find the space of possible motions $G$. We can take the latter to be the maximal subgroup $G^* \subset SE(3)$ of rigid motions under which $\Pi$ is closed[2]

## 4.1 Bilateral symmetry

Bilateral symmetries are generated by the reflection operator $m$ of eq. (2): a pose space $\Pi$ has bilateral symmetry if $H(\Pi) = \{1, m\}$, which induces pairs of symmetric poses $\pi' = m\pi m^{-1}$ as in fig. 2.

Even if poses $\Pi$ are closed under rigid motions (i.e. $G^*\Pi = \Pi$ where $G^* = SE(3)$), in this case there is generally no ambiguity in recovering the object pose from its shape $S$. The reason is that in lemma 4 one has $G^* \cap H(\Pi) = \{1\}$ due to the fact that all transformations in $G^*$ are orientation-preserving whereas $m$ is not. This explains why it is possible to still distinguish left from right sides in most bilaterally-symmetric objects despite symmetries and motions. However, this is not the case for other types of symmetries such as radial.

**Symmetry plane.** Note that, given a pair of symmetric poses $(\pi, \pi')$, $\pi' = m\pi m^{-1}$, the correspondences between the underlying 3D shapes are given by the map $m_\pi : \quad S \to m[S], \quad p \mapsto (m\pi m^{-1}\pi^{-1})(p)$. For example, in fig. 2 this map sends the raised left hand of a person to the lowered left hand in the symmetric pose. Of particular interest are the points where $m_\pi$ coincides with $m$ as they are on the "plane of symmetry". In fact, let $p = \pi(z)$; then:

$$m_\pi(p) = m(p) \quad \Rightarrow \quad m\pi m^{-1}\pi^{-1}(p) = m(p) \quad \Rightarrow \quad m^{-1}(z) = z \quad \Rightarrow \quad z = \begin{bmatrix} 0 \\ z_2 \\ z_3 \end{bmatrix}. \quad (5)$$

## 4.2 Extrinsic symmetries

Our formulation captures the standard notion of extrinsic (standard) symmetries as well. If $H(S) = \{h \in O(3) : h[S] = S\}$ are the extrinsic symmetries of a geometric shape $S$ (say regular pyramid), we can parametrize $S$ using a single pose $\Pi = \{\pi_0\}$ that: (i) generates the shape ($S = \pi_0[\mathbb{S}^2]$) and (ii) has the same symmetries as the latter ($H(\Pi) = H(S)$).

In this case, the pose $\pi_0$ is self-conjugate, in the sense that $\pi_0 = h\pi_0 h^{-1}$ for all $h \in H(\Pi)$. Furthermore, given $S$ it is obviously possible to recover the pose uniquely (since there is only one element in $\Pi$); however, as before ambiguities arise by augmenting poses via rigid motions $G = SE(3)$. In this case, due to lemma 4, if $g\pi_0$ is a possible pose of $S$, so must be $g\pi_0 h^{-1}$. We can rewrite the latter as $(gh^{-1})(h\pi_0 h^{-1}) = (gh^{-1})\pi_0$, which shows that the ambiguous poses are obtained via selected rigid motions $gh^{-1}$ of the reference pose $\pi_0$.

# 5 Learning with ambiguities

In section 3 we have explained how the learning formulation of [38] can be extended in order to learn objects with a bilateral symmetry. The latter is an example where symmetries do not induce an ambiguities in the recovery of the object's pose (the reason is given in section 4.1). Now we consider the case in which symmetries induce a genuine ambiguity in pose recovery.

Recall that ambiguities arise from a non-empty intersection of object symmetries $H(\Pi)$ and object motions $G^*$ (section 4). A typical example may be an object with a finite rotational symmetry group (fig. 3). In this case, it is *not* possible to recover the object pose uniquely from an image, which in turn suggests that $\psi_u(\mathbf{x})$ cannot be learned using the formulation of section 3.

| Method | Eyes | Mouth |
|---|---|---|
| [38] | 23.29 | 15.27 |
| [38] & plane est. | 5.17 | 5.38 |
| Ours | 3.21 | 3.47 |

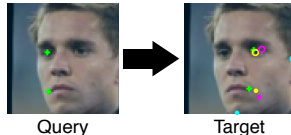

Query — Target

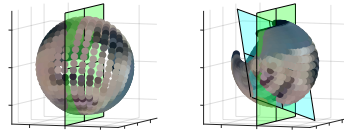

(a) Pixel error when using the reflected descriptor from the left eye or left mouth corner to locate its counterpart on the right side of the face, across 200 images from CelebA (MAFL test subset)

(b) Visualisation of fig. 4a. +: ground truth. ∘, •: [38] with no learned symmetry. ∘, •: [38] with mirroring around the plane estimated using annotations. ∘, •: Our method. Where ∘, • is eye, mouth respectively

(c) Difference between us (left) and [38] (right). We learn an axis aligned frame symmetric around a plane (green), [38] has arbitrary rotation and no guaranteed symmetry plane. But we can estimate a plane using annotations (cyan).

Figure 4: Comparing object frames

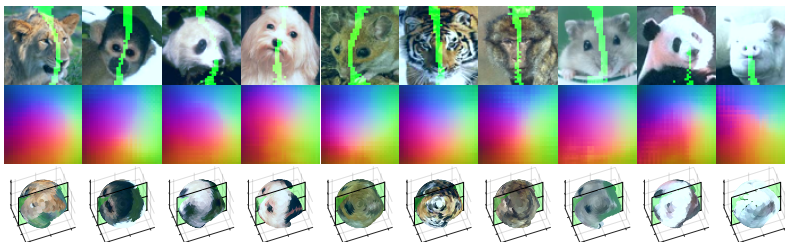

Figure 5: **Bilateral symmetry of animal faces.** The discovered plane of symmetry is shown in green. Top: Inputs, Middle: Colour mapping, Bottom: Embedding (sphere) space

We propose to address this problem by *relaxing* loss (4) in order to discount the ambiguity as follows:

$$\mathcal{L}_{H(\Pi)}(\mathbf{x}, t) = \min_{h \in H(\Pi)} \int_{\Omega} \|v - tu\|_2^{\gamma} p_h(v|u) \, dv du, \quad p_h(v|u) = \frac{\exp\langle h\psi_u(\mathbf{x}), \psi_v(t\mathbf{x})\rangle}{\int \exp\langle h\psi_u(\mathbf{x}), \psi_w(t\mathbf{x})\rangle \, dw} \quad (6)$$

This loss allows $\psi_u(\mathbf{x})$ to estimate the embedding vector $z \in \mathbb{S}^2$ (or $z \in \mathbb{R}^3$) up to an unknown transformation $h$.

## 6 Experiments

We now validate empirically our formulation. To ensure that we have a fair comparison to [38], who introduced learning formulation (4) which our approach extends, we use the same network architecture and hyperparameter values (*e.g.* $\gamma = 0.5$ in eq. (4)). We show that our extension successfully recovers the symmetric structure of bilateral objects (section 6.1) as well as allowing to manage ambiguities arising from symmetries in learning such structures (section 6.2).

### 6.1 Learning objects with bilateral symmetry

In this section, we apply the learning formulation (4) to objects with a bilateral symmetry. Due to the structure imposed on the embedding function by eq. (3), we expect the symmetry plane of the object to be mapped to the plane $z_1 = 0$ in the embedding space (section 4.1). Once the model is learned, this locus can be projected back to an image for visualisation and qualitative assessment. We also test quantitatively the accuracy of the learned geometric embedding in localising object landmarks and their symmetric counterparts.

**Faces.** We evaluate the proposed formulation on faces of humans and animals, which have limited out-of-plane rotations. For humans we use the CelebA [23] face dataset, with over 200K images. We use an identical setup to [38, 39], training on 162K images and employing the MAFL [46] subset of 1000 images as a validation set. For cats we use the Cat Head dataset [45], with 8609 training images. We also combine multiple animals in the same training set, with Animal Faces dataset [36] (20 animal classes, about 100 images per class). We exclude birds and elephants since these images have a significantly different appearance, and add additional cat, dog and human faces [45, 31, 23] (but keep roughly the same distribution of animal classes per batch as the original dataset).

In all cases, we do not use any manual annotation; instead, we use learning formulation (4) using the same synthetic transformations $t \sim \mathcal{T}$ as [38]. Additionally, with 50% probability we also apply a left-to-right flip $m$ to both the image and the embedding space, as prescribed by eq. (4).

Results (figs. 1 and 5) show that our method, like [38], learns a geometric embedding of the object invariant to viewpoint and intra-category changes. In addition, our new formulation localises the intrinsic bilateral symmetry plane in the face images and maps it to a plane of reflection in the embedding space. We note that images are embedded symmetrically with respect to the plane (shown in green in fig. 1, bottom row). The plane can also be projected back to the image and, as predicted by eq. (5), corresponds to our intuitive notion of symmetry plane in faces (fig. 1, top row). Importantly, symmetry here is a statistical concept that applies to the category as a whole; specific face instances need not *be* nor *appear* symmetric — the latter in particular means that faces need not be imaged fronto-parallel for the method to capture their symmetry.

To evaluate the learned symmetry quantitatively we use manual annotations (eyes, mouth corners) to verify if the representation can transport landmarks to their symmetric counterparts. In particular, we take landmarks on the left side of the face (*e.g.* left eye), use $m$ (eq. (3)) to mirror their embedding vectors, backproject those to the image, and compare the resulting positions to the ground-truth symmetric landmark locations (*e.g.* right eye). We report the measured pixel error in fig. 4a. As a baseline, we replace our embedding function with the one from [38] which results in much higher error. This is however expected as the mapping $m$ has no particular meaning in this embedding space; for a fairer comparison, we then explicitly estimate an ad-hoc plane of symmetry defined by the nose, mean of the eyes, and mean of the mouth corners, using 200 training images. This still gives higher error than our method, showing that enforcing symmetry during training leads to a better representation of symmetric objects.

In terms of the accuracy of the geometric embedding as such, we evaluate simply matching annotations between different images and obtain similar error to the embedding of [38] (ours 2.60, theirs 2.59 pixel error on 200 pairs of faces, and both 1.63 error for when the second image is a warped version of the first). Hence representing symmetries does not harm geometric accuracy.

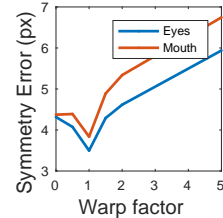

We also examine the influence of the synthetic warp intensity, in fig. 6 we train for 5 epochs scaling the original control point parameters by a factor, indicating we are around the sweet spot and unnatural excessive warping is harmful.

Figure 6: Varying warp intensity

**Synthetic 3D car model.** A challenging problem is capturing bilateral symmetry across out-of-plane rotations. We use a 3D car, animated with random motion [13] for 30K frames. The heading follows a random walk, eventually rotating 360° out of plane. Translation, pitch and roll are sinusoidal. The back of the car is red to easily distinguish from the front. We use consecutive frames for training, with the ground truth optical flow used for $t$ and image size $75 \times 75$. The loss ignores pixels with flow smaller than 0.001, preventing confusion with the solid background. Figure 8 depicts examples from this dataset. Unlike CelebA, the cars are rendered from significantly different views, but our method can successfully localize the bilateral axis accurately.

**Synthetic robot arm model.** We trained our model on videos of a left-right pair of robotics arms, extending the setup of [38] to a system of two arms.

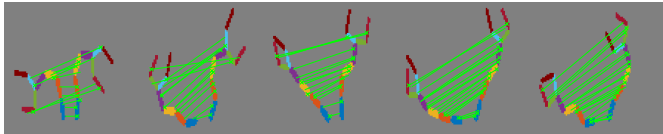

Figure 7: Symmetry in a pair of toy robotics arms

Figure 7 shows the discovered symmetry by joining corresponding points in a few video frames. Note that symmetries are learned automatically from raw videos and ground truth optical flow alone. Note also that none of the images is symmetric in the trivial left-right flip sense due to the object deformations.

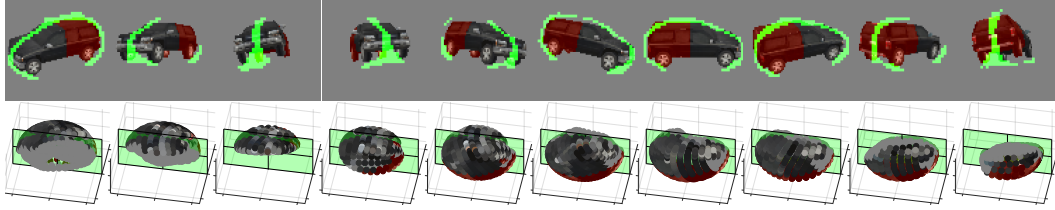

Figure 8: **Bilateral symmetry on synthetic car images**, Top: Input images with the axis of symmetry superimposed (shown in green), Bottom: Image pixels mapped to 3D with the reflection plane (green)

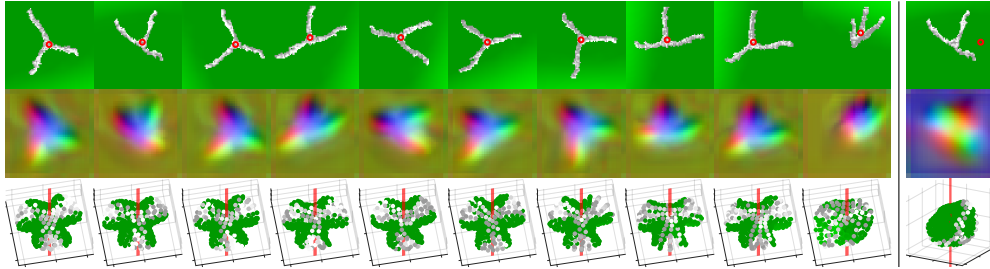

Figure 9: **Rotational symmetry on protein.** Top: Frames, found center of symmetry red. Middle: Colorized object frame, a different colouring is assigned to each leg despite ambiguity. Bottom: Embedding in 3D, it learns to be symmetric around an axis (red). Last column: Without relaxed loss.

## 6.2   Rotational symmetry

We create an example based on 3-fold rotational symmetry in nature, the Clathrin protein [43]. We use the protein mesh[3] and animate it as a soft body in a physics engine [13, 7], generating 200 400-frame sequences. For each we vary the camera rotation, lighting, mesh smoothing and position. The protein is anchored at its centre. We vary the gravity vector to produce varied motion.

We train using the relaxed loss in eq. (6), where $H(\Pi)$ corresponds to rotating our sphere $0°$, $120°$ or $240°$. The mapping then need only be learned up to this rotational ambiguity. As shown in fig. 9, this maps the protein images onto a canonical position which has rotational symmetry around the chosen axis, whereas without the relaxed loss the object frame is not aligned and symmetrical.

We also show results for rotational symmetry in real images, using flower class Stapelia from ImageNet in fig. 10 which has 5-fold rotational symmetry.

## 7   Conclusions

In this paper we have developed a new model of the symmetries of deformable object categories. The main advantage of this approach is that it is flexible and robust enough that it supports learning symmetric objects in an unsupervised manner, from raw images, despite variable viewpoint, deformations, and intra-class variations. We have also characterised ambiguities in pose recovery caused by symmetries and developed a learning formulation that can handle them. Our contributions have been validated empirically, showing that we can learn to represent symmetries robustly on a variety of object categories, while retaining the accuracy of the learned geometric embedding compared to previous approaches.

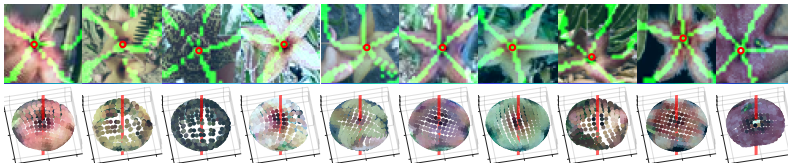

Figure 10: **Rotational symmetry on Stapelia flower.** Superimposed in green, projection into the image of a set of half-planes $72°$ apart in the sphere space. In red, predicted axis of rotational symmetry.

**Acknowledgments:** This work acknowledges the support of the AIMS CDT (EPSRC EP/L015897/1) and ERC 638009-IDIU. We thank Almut Sophia Koepke for feedback and corrections.

## Footnotes

[1]I.e. $\forall p, q \in \mathbb{R}^3 : d(hp, hq) = d(p, q)$.

[2]Being maximal means that $G^*\Pi = G^* \wedge G\Pi = G \Rightarrow G \subset G^*$. The maximal group can be constructed as $G^* = \langle G \subset SE(3) : G\Pi = \Pi \rangle$, where $\subset$ denotes a subgroup and $\langle \cdot \rangle$ the generated subgroup. This definition is well posed: the generated group $G^*$ contains all the other subgroups $G$ so it is maximal; furthermore $G^*\Pi = \Pi$ because, for any pose $\pi \in \Pi$ and finite combination of other group elements, $g_1^{n_1} \ldots g_k^{n_k} \pi \in \Pi$.

[3]https://www.rcsb.org/structure/3LVG

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
