[Supplementary Material · supplementary.pdf]

## Supplementary Material: Modelling and unsupervised learning of symmetric deformable object categories

We show additional qualitative results learning bilateral symmetry on several datasets. Firstly, we try a more challenging setting of the CelebA dataset, by applying rotations with standard deviation of 30 degrees and translations with standard deviation 20% of image width. As shown in fig. 11, our method remains able to learn and recover the axis of symmetry under these conditions.

Secondly, we use an exercise dataset of human pose[1]. Here (fig. 12) the symmetry is recovered accurately with upright pose and certain deformations, but fails in extreme cases.

Finally, we attempt to learn bilateral symmetry on cars, using the CompCars dataset[2]. We observe that, although the symmetry is recovered with frontal images, the plane through the middle of the car seen from side is mistakenly thought to be a symmetry. This is understandable, since we train only using synthetic warps of the same image, so it hard to build up a globally consistent frame. Similarly, the front and back of the car are not disambiguated from each other.

Figure 11: **CelebA trained with large distortions**

Figure 12: **Bilateral symmetry on humans**

Figure 13: **Bilateral symmetry on cars**

[1]Xue, Tianfan and Wu, Jiajun and Bouman, Katherine L and Freeman, William T. Visual Dynamics: Probabilistic Future Frame Synthesis via Cross Convolutional Networks. In NIPS 2016.

[2]Linjie Yang, Ping Luo, Chen Change Loy, Xiaoou Tang. A Large-Scale Car Dataset for Fine-Grained Categorization and Verification, In CVPR 2015

# A   Proofs for Section 4 (Theory)

**Lemma 1.** *The set $H(\Pi)$ is a subgroup of $O(3)$.*

*Proof.* First, note that, since $O(3)$ is the space of extrinsic symmetries of the sphere $\mathbb{S}^2$, then $\mathbb{S}^2 = h\mathbb{S}^2 = h^{-1}\mathbb{S}^2$. This means that the function composition $h\pi h^{-1}$ is well defined. Furthermore, the identity map $h = 1$ is clearly included in $H(\Pi)$, which is therefore not empty. The set is also closed under composition: if $h_1, h_2 \in H(\Pi)$, then using associativity $(h_1 h_2)\pi(h_1 h_2)^{-1} = h_1(h_2 \pi h_2^{-1})h_1^{-1}$ shows that $h_1 h_2 \in H(\Pi)$. It is also closed under inversion: if $h \in H(\Pi)$, then $h^{-1} \in H(\Pi)$ due to the symmetry in the definition.  $\square$

**Lemma 2.** *If $H(\Pi) \subset G$, then $H(\Pi) \subset H(G\Pi)$.*

*Proof.* Let $h \in H(\Pi)$; we need to show that $h \in H(G\Pi)$. To this end, consider the map $r = hgh^{-1}g^{-1}$. We have
$$rg(h\pi h^{-1}) = h(g\pi)h^{-1} \tag{7}$$
By definition, $h\pi h^{-1} \in \Pi$. Furthermore, since $H(\Pi) \subset G$, then $rg = hgh^{-1} \in G$. Hence we conclude that $h(g\pi)h^{-1}$ is contained in $G\Pi$.  $\square$

**Lemma 3.** *$\pi \sim_{H(\Pi)} \pi'$ is an equivalence relation on the space of poses $\Pi$.*

*Proof.* The relation is reflexive because $H(\Pi)$ is a group and thus contain the identity element. It is symmetric because $\pi' = h\pi h^{-1} \Rightarrow \pi = h^{-1}\pi'h$ and $h^{-1} \in H(\Pi)$ as a group is closed under inversion. It is transitive because if $\pi'' = h_2 \pi' h_2^{-1}$ and $\pi' = h_1 \pi h_1^{-1}$ where $h_1, h_2 \in H(\Pi)$, then $\pi'' = h_2 \pi' h_2^{-1} = h_2(h_1 \pi h_1^{-1})h_2^{-1} = (h_2 h_1)\pi(h_2 h_1)-1$ since $h_2 h_1 \in H(\Pi)$ as a transformation group is closed under composition.  $\square$

**Lemma 4.** *Let the pose space $\Pi$ be closed under a transformation group $G$, in the sense that $G\Pi = \Pi$. Then, if pose $\pi \in \Pi$ is a solution of the equation $S = \pi[\mathbb{S}^2]$ and if $h \in H(\Pi) \cap G$, then $\pi h^{-1}$ is another pose that solves the same equation.*

*Proof.* First, note that the composition $\pi h^{-1}$ is always well posed since is any orthogonal transformation $h^{-1} \in O(3)$. Hence the range $h^{-1}\mathbb{S}^2$ of $h^{-1}$ is the same as the domain $\mathbb{S}^2$ of $\pi$. For the same reason, $\pi h^{-1}\mathbb{S}^2 = \pi\mathbb{S}^2 = S$ have the same shape. To conclude the proof, it remains to show that $\pi h^{-1} \in \Pi$. To this end, note that $\pi h^{-1} = h^{-1}(h\pi h^{-1}) = h^{-1}\pi'$. Since $h \in H(\Pi)$, the map $\pi'$ belongs to $\Pi$ by definition of $H(\Pi)$. Since $h \in G$ too, since $\Pi$ is closed to the action of $G$, the map $h^{-1}\pi'$ belongs to $\Pi$ as well.  $\square$