[Reviews · NeurIPS 2018]

Reviewer 1



Summary: This work propose an approach to model symmetries in deformable object categories in an unsupervised manner. This approach has been demonstrated to work for objects with bilateral symmetry (identifying symmetries in human faces using CelebA dataset, cats using cats head dataset, on cars with a synthetic car dataset), and finally for rotational symmetry on a protein structure. Pros: + Overview of the problem and associated challenges. + The proposed approach seems a natural way to establish dense correspondences for non-rigid objects given two views of same object category (say example in Figure-3). In my opinion, correspondences for non-rigid/deformable objects is far more important problem than symmetry (with a potential impact on numerous problems including non rigid 3D reconstruction, wide baseline disparity estimation, human analysis etc). Reading the approach section made me actually think more in terms of correspondences than symmetry. Cons: - While the theory is interesting, I think the current submission is lacking on experiments. Here are my reasons for this statement: (i) The evaluation using CelebA face dataset cannot be termed completely "unsupervised". The face images are aligned using the 6 keypoints (left eye, right eye, nose, left side of mouth, right side of mouth, and chin) and everything is centered around nose. Most of the images have an "inherent" symmetry because of this pre-processing. Therefore, it is hard to tell the role of proposed approach in getting symmetries. I would suggest the authors to use the original CelebA dataset where faces are not aligned and show comparison. (ii) The same argument goes for using Cats-head dataset. I would suggest to instead use Cats and Dogs dataset from Parkhi et al to demonstrate learning of symmetry. In my opinion that would be a sound approach to demonstrate the "unsupervised" part of the approach. I believe that current experiments have a hidden manual supervision and is not completely unsupervised. (iii) This point is a suggestion to make the submission strong -- If the authors agree on (i) and (ii), than I think the most impactful experiment is demonstration of learning symmetries on ImageNet-1k dataset. Use the images from each category, and demonstrate how correspondences and symmetries naturally emerge using this approach without any manual supervision. - An ideal comparison of this approach would be the evaluation on synthetic humans dataset (as done by Raviv et al [27]). The authors have provided a qualitative analysis in the supplementary material for synthetic humans but an extensive analysis with human study is the one that I would have loved to see. Even more interesting would have been the use of data from L. Wei et al (CVPR 2016) < http://www.hao-li.com/publications/papers/cvpr2016DHBCUCN.pdf > for demonstrating symmetries, and to verify the hypothesis of lemma-3, lemma-4, and bilateral symmetry. Suggestion: 1. The proposed approach has a flavor of estimating symmetry or similarity via composition, so it would be good to include those work in Section-2 specifically the ones by Boiman and Irani (NIPS 2006, ICCV 2005).

Reviewer 2



The paper is a straightforward extension of Thewlis et al [31] to do unsupervised learning of symmetries. The main contribution is to realize that the formulation in [31] applies for the problem. This is achieved by a rather trivial addition of "reflection flip" m to to the loss from Eq 5 in [31]. The experimental results show that this recovers reasonable models of symmetry from images for simpler objects like faces and vehicle with limited transformations - and that harder cases are still potentially challenging. The paper also includes a set of lemmas on object symmetry in the presence of transformations, which appear pretty straightforward. I found that the paper is hard to read -- it is heavy on group theory notation early, while lacking details on a practical instantiation of the method, which makes it hard to follow the exposition and to understand the differences from [31]. Ultimately, the paper lacks concrete method implementation details, requiring a careful read of [31]. Examples of lack of clarity include: -- Network architecture is not described. Training details are missing. -- In Eq 6, how is min H(П) selected. How does one compute / approximate the integrals exactly over du and dw. -- How are areas of the training images that don't contain the object embedded / handled? -- In equation 6, should the second (diversity loss) by p(v | u,h) ? -- In line 208, mgx is mentioned but Eq (5) contains mtx. In Equation 6, if one does not explicitly enumerate a discrete set of rotations (as it appears to be done for the protein case), is it still possible to recover the symmetry? There's a fairly rich literature on detecting symmetries in 3D, potentially more work can be cited: -- Cluttered scene segmentation using the symmetry constraint, A. Ecins et al, ICRA 2016 -- Shape from Symmetry Sebastian Thrun, Ben Wegbreit: ICCV 2005

Reviewer 3



This work focuses on learning the structure of categories from raw images of symmetric objects. Their approach focuses on symmetries in object deformations rather than that in the geometric shapes. They attempt to extend the object frame model to capture symmetries. They claim that their approach can provide reasonings regarding ambiguities that can occur while recovering the pose of a symmetric object. The authors study an interesting problem where instead of studying symmetries in shape they study the symmetry in object categories. They claim that intrinsically symmetric objects might not appear symmetric due to different pose/point of view etc. Therefore, studying symmetries in object deformations may be helpful. Instead of modeling object categories as shapes, poses etc., they model them as embedding a sphere into R^3. Their notion of symmetry as a subset of linear isometries is also interesting. Their approach to ambiguous poses as rigid shifts of reference poses is also interesting. They also demonstrate empirically that their approach can work better than their main baseline. I am curious about the influence of the warp distribution T. Any thoughts about different warps? Also, the images would be much clearer if they were made bigger and higher resolution. Furthermore, what sort of transformations h can be easily handled by the proposed approach (Eq 6). The authors seem to provide an example that was synthetically created based on rotational symmetry in nature . It would be interesting to know how the system might perform in non-synthetic scenario and what issues it might face.